# LARGE LANGUAGE MODELS CAN BE MORE ROBUST MULTIPLE CHOICE SELECTORS THROUGH ATTENTION INTERVENTION

## ABSTRACT

Multiple-choice question (MCQ) is a common task for evaluating large language models (LLMs). LLMs' performance on MCQ is often affected by various biases. Previous research has extensively examined the impact of **inherent option bias** on MCQ predictions, where this bias refers to a preference for a specific option ID token introduced during the model's training. However, in an in-context learning scenario, few-shot prompting can also introduce a form of bias, known as **context option bias**. This occurs, for instance, in extreme cases where all demonstration answers are consistently option A, in which case LLMs may predict A for the given question whatever the question is. Context option bias can significantly degrade LLMs' performance. To better observe the LLMs' behavior when affected by the context option bias, we deliberately use demonstrations with obvious context option bias for MCQ to amplify the effect. The results indicate that certain attention heads in LLMs are particularly sensitive to context option bias. Motivated by this observation, we propose our approach, CoLo, to address this issue. First, using samples with ordinary and biased demonstrations as input, CoLo **co**mpares the outputs of two types of inputs and **lo**calizes attention heads sensitive to context option bias through sequential interventions. Then, we propose an attention scaling-based method to intervene in the identified attention heads during the inference stage, thereby mitigating the impact of context option bias on the LLMs' predictions. Experimental results demonstrate that CoLo effectively alleviates the impact of context option bias and improves the LLM's robustness on MCQ tasks.

## 1 INTRODUCTION

Multiple-choice question (MCQ) is a common type of question-and-answer format in daily life and is also a common method used in the field of natural language processing to test the generalization ability of large language models (LLMs). There exist benchmarks and datasets specifically designed, spanning a variety of fields(Hendrycks et al., 2021; Talmor et al., 2018; Clark et al., 2018). We hope LLMs can well understand the question and choose the most appropriate answer from all options. For this purpose, many efforts have been made and few-shot in-context learning has been shown to be one of the effective methods. It provides some demonstrations before the question as prompts and can largely improve an LLM's performance on MCQs.

Despite the above efforts, large language models are still often affected by certain *biases* and yield unexpected answers. Previous works(Wang et al., 2023; Pezeshkpour & Hruschka, 2023; Zheng et al., 2023) have investigated several **inherent option bias** of LLMs. For example, experiments indicate that `gpt-3.5-turbo` tends to prefer option A (Zheng et al., 2023), which may result from the uneven distribution of options in the training corpus. However, bias introduced by context, referred to as **context option bias**, has not been carefully studied. As shown in Figure 1, take questions with only two options as an example, if all demonstrations' answers happen to be A, then LLMs will also prefer to predict A no matter what question is given. In contrast, if answers for demonstrations do not show too much preference to a certain option, the model will be more inclined to choose the correct answer, *i.e.*, B in this example. The results indicate that context option bias has an impact on

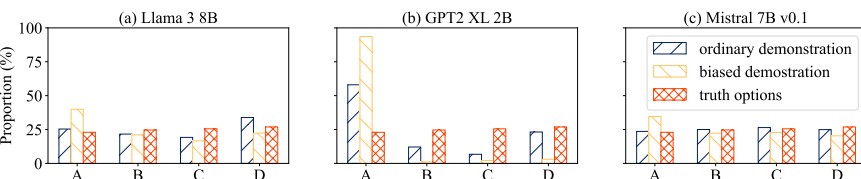

Figure 1: The context gives the demonstration and question, allowing the model to predict the correct option. Due to the bias in the demonstration, it will affect the final prediction result.

Figure 2: Biased context is constructed by few shot prompts through answer is always *A*. We counted the predictions of some language models in biased and ordinary demonstrations respectively, as well as the proportion of option distribution of real labels on MMLU dataset(Hendrycks et al., 2021)

an LLM's prediction. Figure 2 provides another evidence, that is, when given demonstrations all with answers A, LLMs will predict much more A than with ordinary demonstrations[1].

However, demonstrations with evenly distributed options still introduce context option bias towards certain answer choices. We observed that demonstrations with evenly distributed answer options significantly influence the model's predictions. For instance, in the 5-shot Gemma-2B model, we modified the sequence of options in the MMLU dataset demonstrations to two different configurations: A-B-C-D-A and D-C-B-A-D. Compared to the random sequence demonstration, the total number of A selections in configuration A-B-C-D-A decreased by 33%, while the total number of D selections in configuration D-C-B-A-D decreased by 47%. We speculate that this behavior arises because the model tends to avoid selecting the same option consecutively, believing that the probability of repeated occurrences of the same option is low, thus introducing a bias in answering questions with similar option distributions.

To better excavate the reasons behind this problem and resolve it, we deliberately construct some demonstrations with obvious context option bias and compare the model's behavior under these conditions with that under ordinary demonstrations. The experiments find that even with simple option swaps in demonstrations, the attention map's distribution in an LLM varies significantly. This observation inspires us to test the attention distribution under conditions with obvious biased demonstrations, identify the attention heads that are more sensitive to context option bias, and intervene accordingly to mitigate the impact of context option bias on the model behavior.

Following this motivation, we propose our method CoLo to localize and mitigate context option bias in LLMs. Particularly, (1) CoLo first takes samples with biased demonstrations as input to obtain the LLM's predictions, which we refer to as biased answers. (2) We then intervene in different layers and different attention heads within each layer. If the LLM's predictions after the intervention is more inclined towards the correct or ordinary answers, it indicates that the corresponding attention head is sensitive to context option bias and likely to lead the model to biased answers. (3) CoLo selects a set of attention heads that are most sensitive to context option bias and applies interventions to these heads simultaneously during the subsequent inference stage.

Extensive experiments show that for MCQ in an in-context learning scenario, when there is context option bias in demonstrations, our method can effectively mitigate the impact of the bias, improving an LLMs' accuracy on MCQ to a level similar to that with ordinary demonstrations without affecting other LLMs' abilities. Besides, the identified attention heads demonstrate powerful generalization across datasets and tasks. That is, for a certain model, the attention heads identified with one dataset also applies to other datasets. More importantly, even when the demonstrations look unbiased, applying our proposed intervention can also further enhance the model's accuracy. We assume it

---

[1]Ordinary demonstrations mean those with the same questions as biased demonstrations but do not show preference to a certain option. We implement this by swapping options in demonstration questions.

is because seemingly unbiased demonstrations may also contain some context option bias that are imperceptible, thereby affecting the model's predictions.

It is worth summarizing our contributions as follows:

- We propose a method to identify attention heads that are sensitive to context option bias in multiple-choice questions under an in-context learning scenario.
- Our attention scaling-based method mitigates the impact of context option bias on the LLM's predictions and improves performance without compromising other capabilities.
- Extensive experiments are conducted and the results show the effectiveness of our method.

## 2 RELATED WORK

**Inference intervention techniques**. Inference intervention comprises various methods designed to modulate the behavior of large-scale models post-training. Commonly employed inference intervention strategies include activation editing(Li et al., 2024b), weight editing(Dai et al., 2022; Meng et al., 2022), guidance vectors(Zou et al., 2023), and alter the output distribution through comparison(Li et al., 2022; Chuang et al., 2023). Our method focuses on modifying the attention distribution by attention scaling, which is a technique that originates from GPT-2(Radford et al., 2019). Contrary to the broader method of global attention scaling of GPT-2, our approach is distinctly less invasive, selectively targeting attention heads that are associated with context option bias. By identifying and scaling the attention heads linked to context option bias, our method effectively mitigates this bias using a significantly smaller dataset, requiring far fewer examples than those needed for reinforcement learning (RL) (Ouyang et al., 2022) and fine-tuning-based methods (Hu et al., 2021). Our approach to structuring the comparison of specific contexts is analogous to the ICD method(Zhang et al., 2024).

**Bias of LLMs**. Our study uses bias to refer to the production of systematic errors in a model. Although In-context learning can bring considerable task learning capabilities to the model(Pan, 2023), but Turpin et al. (2024) assess the LLMs' unfaithfulness to CoT interpretation by constructing context option bias features. That mean the abilities brought by in-context learning are not necessarily useful. Pezeshkpour & Hruschka (2023) point out the sensitivity of option position to model multiple-choice questions and Zheng et al. (2023) suggest that the option bias of the model comes more from the option token bias, and proposed Pride, a method to alleviate the option prior bias by estimating the prior and applying prior debiasing. Distinct from the Pride, CoLo intervenes internally within the model to mitigate context option bias without impacting text generation processes, such as Chain of Thought (CoT)(Kojima et al., 2022). This technique requires only a minimal number of samples to identify a general head that responds to biases across various contexts. In contrast, Pride functions primarily as a pre-processing technology, necessitating recalibration of bias each time the context changes, and it often performs suboptimally in scenarios with limited data samples.

## 3 LOCALIZING AND MITIGATING CONTEXT OPTION BIAS

### 3.1 OVERVIEW

During the model's inference, different attention heads perform distinct functions(Zhang & Nanda, 2024). While some attention heads may correspond to option bias, it is not feasible to directly identify which ones specifically linked to option bias during inference.

To identify attention heads strongly associated with option bias, it is necessary to amplify the presence of the bias. Due to the difficulty of directly manipulating the model's inherent option bias, we choose to amplify context option bias by constructing biased demonstrations. We can get the state output of the model in these two different situations $state_b$ and $state_o$ corresponding to biased and ordinary demonstrations respectively. The gap between the two states arises from the amplification of context option bias. We employ an intervention function $f$ that acts on specific attention heads to reduce the state difference between $state_b$ and $state_o$. The extent of this reduction is compared to evaluate the correlation strength between these attention heads and context option bias, allowing us to select the most relevant set of attention heads. The algorithm flow is described in detail in Section 3.2

After identifying attention heads strongly associated with context option bias, different intervention methods are employed to modify the attention distribution of these heads and mitigate context option bias. We describe the intervention methods utilized in Section 3.3.

## 3.2 LOCALIZING ATTENTION HEADS

We amplify context option bias by swapping options in the demonstration. Specifically, we consolidate all correct answers into a single option, such as always setting the correct answer to A. This creates a biased demonstration, denoted as $d_b$, while the ordinary demonstration is $d_o$, and the problem is represented by $q$. Each sample includes the true label $y_t$. We randomly select $N$ samples to create the set used for localizing attention heads, denoted as $\mathcal{D} = (d_o^i, d_b^i, q^i, y_t^i)_{i=1}^N$.

We describe the general pattern by which large models accomplish selection tasks as:

$$p(y|d, q) = \text{softmax}(\{\phi(g)_c, c \in \mathcal{C}\}), \tag{1}$$

where $\mathcal{C}$ represents the token ID of the option after tokenizer encoding, the vocabulary head $\phi(\cdot)$ predicts the probability of the choice token, $g$ is the output of the last layer of the model.

The probability of the correct option of the model under biased demonstration is $p(y_t|d_b, q)$, and under ordinary demonstration is $p(y_t|d_o, q)$. Choose the option with the highest probability as the answer $y$:

$$y = y_c, \underset{c}{\arg\max} \, p(y_c|d, q), \tag{2}$$

thus, we can obtain $y_o$ through $d_o$. The probability of the model output option after intervention function $f$ will change. The probability distribution after intervention function $f$ is recorded as $p'(y|d, q)$. Particularly, $p_l'^{th}(y|d, q)$ represents probability distribution after intervention function $f$ act on the $h$-th attention head in the $l$-th transformer decoder block.

We define $E_f$ to quantify the effectiveness of the intervention function, described as:

$$E_f = p'(y|d_b, q) - p(y|d_b, q), y \in \{y_o, y_t\}, \tag{3}$$

where $y \in \{y_o, y_t\}$ represents our expectation that, following the application of the intervention function, the model's output previously influenced by biased demonstrations will either align with the label $y_o$, corresponding to the output under ordinary demonstrations, or with the true label $y_t$ of the current sample. Another intuitive approach is to use KL divergence to define the effectiveness of the intervention. To this end, we attempted to measure it by calculating the KL divergence between the model's output after the intervention under biased demonstrations and the output from the original, unbiased demonstration, as well as the true label. However, this approach yielded poor results, as detailed in the Appendix G.

By further partitioning the sample dataset $\mathcal{D}$ and conducting multiple rounds of voting, the top $K_l$ layers are selected, followed by the selection of the top $K_h$ heads within each layer based on their $E_f$ values. The details of the localizing algorithm are provided in Algorithm 1. The rationale for partitioning $\mathcal{D}$ and performing multiple rounds of voting is to mitigate the influence of outliers.

It is essential to emphasize that we first identify the top $K_l$ layers and then select the top $K_h$ heads within these layers. This approach may overlook some heads in lower-ranked layers that could be of greater importance. Although traversing all heads would produce better results, the computational cost is significantly higher. Therefore, we adopt this compromise method to efficiently localize attention heads.

We employ random sampling to partition the sample set and use multiple rounds of voting to identify the top $K_l$ layers. The context option bias present in each sample set is determined during this process. If the identified attention head corresponds to the option bias, its focus should be concentrated on a specific layer and remain robust across samples containing the same context option bias but differing in content. We use the indicator $S \in [0, 1]$ to measure the stability of the identified attention head:

$$S = \sum \frac{cnt_l}{m \times K_l}, l \in \mathcal{L}, \tag{4}$$

The variable $cnt$ represents the number of selections after $n$ rounds of sampling, while $m$ denotes the number of samples in each round. A larger value of $S$ indicates greater concentration and higher effectiveness of the attention head localization strategy.

---

**Algorithm 1** Localizing Attention Heads

---

**Require:** Language model, test samples $\mathcal{D} = \{(d_o^i, d_b^i, q^i, y_t^i)\}_{i=1}^N$, rounds number $n$, sample number of every round $m = N/n$, divide $\mathcal{D}$ into $\{\mathcal{D}_i\}_{i=1}^n$.
**Ensure:** Model attention heads set to intervene $\mathcal{H}$
 1: Initialize decoder layers set $\mathcal{L} = \varnothing$ and the attention heads set $\mathcal{H} = \varnothing$      ▷ Initialization
 2: Sample the estimation samples $\mathcal{D}_e$ under $K$ and the remaining samples $\mathcal{D}_r = \mathcal{D}\backslash\mathcal{D}_e$
 3: **for** $\mathcal{D}_i \in \{\mathcal{D}_i\}_{i=1}^n$ **do**
 4:     **for** $l \in$ model.layers **do**
 5:        **for** $(d_0, d_b, q, y_0) \in D_i$ **do**
 6:           $score_l += p_l'(y_o|d_b, q) + p_l'(y_t|d_b, q)$
 7:        **end for**
 8:        select top $K_l$ layers $l_1, \ldots, l_{K_l}$ of $score_l$
 9:     **end for**
10:     Find the top $K_l$ layers with the largest scores and add them into $\mathcal{L}$    ▷ Get intervene layers
11:     $cnt_{l_i} + +, i = 1, \ldots, K_l$
12: **end for**
13: select top $K_l$ layers in $cnt$ get $\mathcal{L}$
14: **for** $l \in \mathcal{L}$ **do**
15:     **for** $h$ in l.heads **do**
16:        **for** $(d_0, d_b, q, y_t) \in D$ **do**
17:           $score_l^h += p_l'^h(y_o|d, q) + p_l'^h(y_t|d_b, q)$
18:        **end for**
19:        select Top $K_h$ of $score_l^h$ ,add $(l, h)$ to $\mathcal{H}$
20:     **end for**
21: **end for**
22: **return** $\mathcal{H}$

---

### 3.3 INTERVENTION METHODS

To establish the notation and context, we briefly outline some fundamental aspects of transformer architecture(Vaswani et al., 2017), a sequence of transformer layers indexed by the variable $l$.

Throughout the inference process, the token undergoes initial encoding into the embedding space via the embedding layer $x_0 \in \mathbb{R}^{DH}$, initiating the residual stream. We use $h$ to represent head index of each layer. The inputs of multi-head attention(MHA) are $Q_h \in \mathbb{R}^D$, $K_h \in \mathbb{R}^D$, $V_h \in \mathbb{R}^D$, all represented by $x_i$ is obtained through linear operation. In each layer, MHA consists of $H$ heads. Each head is an independent linear operation. After concating the $H$ attention heads, it is processed through $W_o \in \mathbb{R}^{DH \times DH}$ projection gets the result of Multi-head Attention.

$$x_{l+1} = x_l + \text{concat}(head_0^l, \ldots, head_{H-1}^l)W_o \tag{5}$$

Every attention head can be written as:

$$head_h^l = Attn(Q_h^l, K_h^l, V_h^l) = \text{softmax}(f(\frac{Q_h^l K_h^l}{\sqrt{D}})) \times V_h^l, \tag{6}$$

Specifically, in the standard transformer, $f(x) = x$, whereas in our approach, the function $f$ represents an intervention method applied to the attention weights. After preliminary experimental attempt in Appendix D, we guess that attention scaling can better adjust the distribution of attention weight thereby reducing the gap in the final output state of the model. We conducted further experiment with different forms of $f(x)$, such as scaling and zeroing, on specific heads to modify the distribution of attention weights in order to achieve the highest possible $E_f$:

$$f(x) = \begin{cases} x/T, & scaling \\ 0, & setting\ zero \\ \text{mean}(x), & setting\ mean\ value \end{cases} \tag{7}$$

Attention scaling is equivalent to adding the temperature coefficient T to the softmax of the attention score. Given the relatively limited exploration of the softmax operation of attention scores in prior work, LLMs conventionally set the temperature parameter to 1 during reasoning.

Table 1: Compare different intervention methods $f$ on `Gemma-2B` and evaluate on MMLU. Report accuracy improvement $\delta$ and localizing stability $S$. $b$=0.5 for translation.

| Methods | Scaling(T=0.5) | Scaling(T=2) | Setting zero | Setting mean value |
|---|---|---|---|---|
| $S$ | 0.8 | 0.25 | 0.23 | 0.35 |
| biased $\delta$(%) | 2.22 | -0.26 | -0.19 | 0.14 |
| ordinary $\delta$(%) | 1.74 | 0.03 | -0.08 | 0.15 |

After identifying the head set $\mathcal{H}$ through Algorithm 1, we intervene in these attention heads selected during model inference to obtain the prediction results.

$$head_h^l = \mathrm{softmax}(f(\frac{Q_h^l K_h^l}{\sqrt{D}})) \times V_h^l, (l, h) \in \mathcal{H} \qquad (8)$$

### 3.4 DISCUSSION

**How many additional labeled samples are required for CoLo?** Small amount of additional sample are required in Algorithm 1. Moreover, we can expand the dataset by modifying the context option bias in the demonstration. For instance, in a four-option MCQ, we can alter the options to consist entirely of `A`/`B`/`C`/`D`, thereby increasing the sample size fourfold. Clearly, a greater number of additional samples enhances the stability of the final attention head. We ultimately decided to use only 10 additional samples and expanded the dataset to $N$=40 by experiment.

**How much additional computational overhead does CoLo introduce?** The additional computational overhead introduced by the localization process is proportional to the number of samples $N$ used for positioning. In our experiment, the model requires approximately $150N$ additional inferences, which depends on the number of layers and attention heads in the model. The calculation method is detailed in Appendix E. However, once the relevant attention heads are identified, it can be applied to other datasets without incurring any further computational overhead during inference.

## 4 EXPERIMENTS

In our experiments, we employed LLMs such as Llama(Touvron et al., 2023) and Gemma(Team et al., 2024). Our methodology is also applicable to other LLMs with accessible internal attention weights and computational mechanisms.

### 4.1 MAIN RESULTS

We select the MMLU (Hendrycks et al., 2021) as our benchmark. Initially, we randomly select 10 samples from this dataset and increase the sample size fourfold to construct $\mathcal{D}$. Utilizing this set, we identify the attention heads by localizing algorithm and then perform intervention, subsequently assess accuracy on the remaining samples.

According to the localizing method we gave, we try different attention intervention functions and find that using attention scaling method has the best effect as shown in Table 1, which is reflected in the most obvious improvement in accuracy on the dataset and higher localizing stability $S$. This is the same as our guess. The following will provide a detailed introduction to the effects of using attention scaling method as intervention function $f$ on large-scale datasets.

Given that the number of samples selected for the experiment is considerably smaller than the total available, the attention heads identified for each sample are not identical, though they exhibit significant similarity. This procedure is replicated five times to mitigate random variability, and we report the mean accuracy achieved across these iterations.

To quantify the mitigation of option bias, we use recall standard deviation (RStd) as an indicator(Zheng et al., 2023), measuring the balance of recall rates across different option IDs. Additionally, we compare the bias mitigation performance and computational overhead with PriDe. Since CoLo operates during inference, it can be combined with PriDe to further reduce option bias.

Table 2: The experimental results of CoLo on the MMLU dataset, along with a comparison to Pride, encompass evaluations of computational cost, accuracy, and RStd performance. Biased 5 shot is constructed by consistently modifying the standard answers to A.

| Model | Cost | Zero Shot | | Ordinary (5 shot) | | Biased (5 shot) | |
|---|---|---|---|---|---|---|---|
| | | Acc(%) | RStd(%) | Acc(%) | RStd(%) | Acc(%) | RStd(%) |
| Gemma-2B | 1 | 33.8 | 21.8 | 39.9 | 14.3 | 38.2 | 16.1 |
| +CoLo | ×1.19 | 36.2 | 12.6 | 41.7 | 10.8 | 40.5 | 7.0 |
| +Pride(5%) | ×1.15 | 34.1 | 10.6 | 41.7 | 7.5 | / | / |
| +CoLo Pride(5%) | ×1.34 | **37.0** | **5.0** | **42.5** | **4.2** | / | / |
| Gemma-7B | 1 | 61.6 | 5.2 | 62.6 | 5.4 | 59.3 | 10.6 |
| +CoLo | ×1.29 | 62.0 | 5.5 | 63.4 | 3.7 | 61.5 | 10.9 |
| +PriDe(5%) | ×1.15 | 61.6 | 5.2 | 64.4 | 5.0 | / | / |
| +CoLo PriDe(5%) | ×1.44 | **62.1** | **1.8** | **64.5** | 3.5 | / | / |
| Llama2-7B | 1 | 40.9 | 13.9 | 45.5 | 10.7 | 43.5 | 11.5 |
| +CoLo | ×1.45 | 43.2 | 4.4 | 46.5 | **3.8** | 45.6 | 6.2 |
| +PriDe(5%) | ×1.15 | 40.0 | 5.7 | 45.6 | 7.0 | / | / |
| +CoLo PriDe(5%) | ×1.60 | **43.5** | **4.2** | **46.8** | 4.2 | / | / |
| Llama3-8B | 1 | 61.4 | 13.8 | 64.6 | 7.8 | 60.7 | 12.0 |
| +CoLo | ×1.45 | 61.9 | 6.8 | 64.8 | 2.3 | 62.0 | 8.9 |
| +PriDe(5%) | ×1.15 | 63.6 | 5.3 | **65.0** | 2.1 | / | / |
| +CoLo PriDe(5%) | ×1.60 | 63.5 | 3.9 | 64.9 | **1.9** | / | / |

Based on the experimental results in the Table 2, we can conclude that CoLo effectively mitigates option bias, particularly in addressing context option bias. The reduction in RStd indicates a weakening of the model's option bias. Although our primary objective was to reduce option bias, the simultaneous improvement in model accuracy alongside the reduction in RStd further validates the effectiveness of the CoLo. Moreover, the biased demonstration was specifically designed to amplify context option bias. Under these conditions, CoLo demonstrates greater effectiveness in reducing RStd and improving accuracy compared to ordinary demonstrations, indicating its ability to successfully alleviate context option bias.

CoLo alleviates option bias under both ordinary demonstration and zero-shot settings. A plausible explanation is that, while the option distribution in the few-shot setting is uniform, it still contains implicit context option bias as shown in Appendix C. Furthermore, the mitigation of option bias in the zero-shot setting may stem from the inclusion of the question as part of the input context, which can introduce additional context option bias. In summary, the reduction of bias and improvement in accuracy observed in both ordinary demonstration and zero-shot scenarios further underscore the significance and broad applicability of the CoLo.

As shown in the Table 2, we compare the performance of CoLo and PriDe on the MMLU dataset and find that combining the two methods achieves a more effective debiasing outcome. Additionally, we compare CoLo and PriDe across different settings, as well as in combination, and present detailed experimental results on domain transfer within MMLU dataset. The results in Appendix H demonstrate that CoLo exhibits distinct advantages in domain transfer.

## 4.2 ROBUSTNESS OF DIFFERENT EXPERIMENT SETTINGS

To validate whether the attention heads obtained through CoLo genuinely correspond to context option bias, we perform **mode transformations** on the original MCQ input format, varying factors such as the number of options, the length of demonstrations, and the format of option identifiers. Through these experiments, we aim to verify the following two points:

1. Whether the attention heads selected in the original MMLU remain useful to mitigate context option bias after the mode transformations.
2. Whether the attention heads selected in different modes is consistent.

For the first point, we selected the model's attention heads based on the sample set $\mathcal{D}$ in the base mode and reported the accuracy improvements across different modes of $\mathcal{D}$. As shown in Figure 3(a),

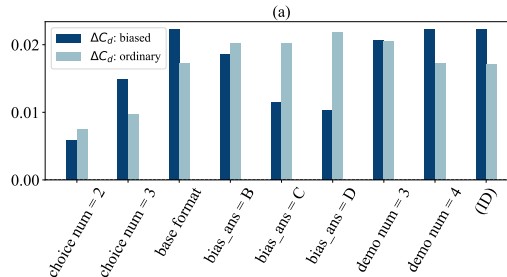 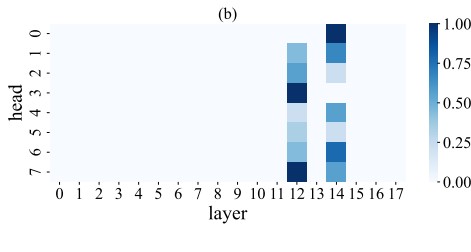

Figure 3: (a) Starting with localizing attention head in the base mode (where the number of choices is 5, the number of demonstrations is 4, and the biased answer is A), we modify the mode for different samples and report the resulting accuracy improvements. (b) We localize attention heads across different modes and present the localization frequency for each attention head. The experimental results are based on the `Gemma-2B` model applied to the MMLU dataset.

Table 3: Cross-dataset experiments were conducted **using the attention heads obtained from MMLU**, along with ordinary few-shot learning across all datasets. For TruthfulQA, the MC1/2/3 indicators were used for evaluation, while accuracy was employed as the evaluation metric for the remaining datasets. FS stands for 5-shot in table. We use lm-eval-harnessGao et al. (2024) as the evaluation tool.

| Model | MMLU | | CMMLU | | CEVAL | | AGIEVAL | TruthfulQA | | |
|---|---|---|---|---|---|---|---|---|---|---|
| | ZS | FS | ZS | FS | ZS | FS | ZS | MC1 | MC2 | MC3 |
| Gemma-2B | 33.0 | 41.7 | 28.4 | 30.9 | 26.5 | 31.3 | 31.0 | 0.233 | 0.371 | 0.173 |
| +CoLo | 36.2 | 42.8 | 29.8 | 31.7 | 29.8 | 32.2 | 31.6 | 0.241 | 0.372 | 0.173 |
| Gemma-7B | 61.6 | 64.5 | 44.8 | 48.9 | 41.2 | 48.2 | 38.7 | 0.308 | 0.476 | 0.228 |
| +CoLo | 62.0 | 65.2 | 45.4 | 49.8 | 44.4 | 49.4 | 38.8 | 0.322 | 0.479 | 0.236 |
| Llama2-7B | 40.9 | 45.6 | 27.2 | 32.7 | 30.0 | 34.0 | 32.3 | 0.286 | 0.434 | 0.207 |
| +CoLo | 43.2 | 46.8 | 28.2 | 32.8 | 30.3 | 35.0 | 33.1 | 0.291 | 0.440 | 0.215 |
| Llama3-8B | 61.4 | 66.5 | 47.5 | 50.5 | 47.9 | 51.9 | 37.3 | 0.324 | 0.492 | 0.244 |
| +CoLo | 61.9 | 66.9 | 47.2 | 50.6 | 49.7 | 52.0 | 39.0 | 0.321 | 0.491 | 0.245 |

the results indicate that accuracy improves following mode transformation. For the second point is to use different $\mathcal{D}$ after mode transformation to select attention heads. For different modes, the heads selected are similar in Figure 3(b). The result indicates that the attention heads selected through our method is robust to mode transformations.

### 4.3 CROSS-DATASET GENERALIZATION

In order to further illustrate that the head obtained by positioning corresponds to the context option bias of the model, and has a certain degree of robustness, we will apply the attention head obtained through random sampling localizing in MMLU and the intervention method of attention scaling to different fields and different forms tasks , 1) MCQ dataset CMMLU(Li et al., 2024a) and CEVAL(Huang et al., 2023), 2) comprehensive dataset AGIEVAL(Zhong et al., 2023), including multiple choice questions and cloze, 3) TruthfulQA, MCQ dataset but not select specific option IDs.

The results of the experiment in Table 3 illustrates the improvement in accuracy of MCQ in other fields, which demonstrates the robustness of our approach in mitigating option bias across datasets. Additionally, the stable performance on the TruthfulQA indicates that our intervention method will not cause the model to lose the ability to generate general text.

### 4.4 INTERVENTION PARAMETERS

An important parameter $T \in \mathbb{R}^+$ is involved in CoLo. Although no theoretical framework exists to determine its optimal value, we explore its effects through experimental analysis.

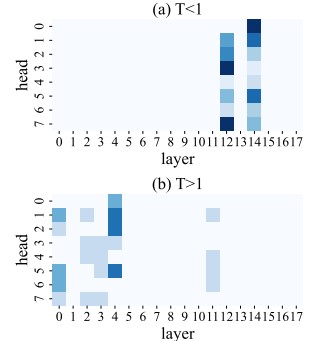 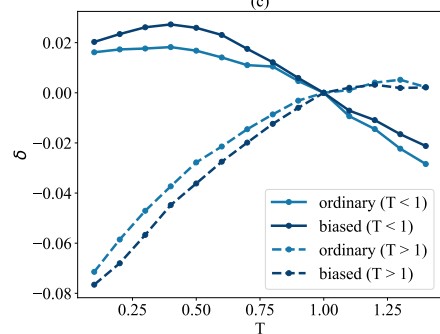

Figure 4: (a) and (b) test various T values for T<1 and T>1, respectively, and observe the frequency of the localized head. (c) Different attention scaling coefficients T were tested on `Gemma-2B` using the head identified by CoLo, and the accuracy improvements were evaluated on the MMLU.

We found that for T < 1, the positions of the heads are similar, while for T>1, the positions become more dispersed, as shown in Figure 4(a) and (b). Additionally, we conducted experiments to vary the value of T after determining the head positions for T < 1 and T > 1. From Figure 4, we observed that increasing T beyond 1 has no effect on improving accuracy. We hypothesize that reducing the variation in the attention distribution does not impact the model's predictive performance. In this case, T>1 may have identified a set of heads that are not critical for predicting the final outcome. Aligning the attention distribution closer to the mean appears to have no significant influence on the results.

As long as T is within a reasonable interval, T∈[0.1, 0.6], there is no significant difference in the final bias mitigation effect and accuracy improvement.

### 4.5 EXPLANATIONS AND ANALYSES

We give reasonable explanations why our method can mitigate context option bias from model's focus changes after intervention.

We define the focus coefficient to measure the extent to which the model attends to different parts of the context, including context demonstrations $C_d$, question $C_q$, and self-rational $C_r$. $p_l$ represents the last position of the encoded text, $p_d$ represents the end position of the ID after the demonstration is encoded by the tokenizer, $p_q$ represents the end position of the question. According to the decoder structure, we can calculate its attention coefficient for each attention head when the model finally determines the answer to the question. We use $A_{l,h}(i,j)$ represents the attention score of the $(i,j)$ position of the $h$-th attention head in the $l$-th layer. After normalizing, the final attention coefficient $C_d$, $C_q$ and $C_r$ is obtained.

$$C = \sum_{j=s}^{e} A_{l,h}(p_l, j)/|e - s + 1|, (s,e) \in \{(0, p_d), (p_d + 1, p_q), (p_q + 1, p_l)\} \tag{9}$$

**Options with high uncertainty are more susceptible to context option bias in demonstrations**. In Table 4, context option bias is intensified by constructing biased demonstrations, and MCQs are divided into two categories: "doubtful," where selections change after switching from ordinary to biased demonstrations, and "firm," where selections remain unchanged. The confidence in the altered options is significantly lower than in the unchanged ones, averaging 0.25 for the four-option questions. Thus, when context option bias is amplified, options with lower certainty are more likely to change.

Table 4: The proportion of doubtful and firm types in the MMLU, and the confidence to the question.

| Model | proportion % | | confidence | |
|---|---|---|---|---|
| | firm | doubtful | firm | doubtful |
| `Llama2-7B` | 79.3 | 20.7 | 0.402 | 0.269 |
| `Gemma-2B` | 73.8 | 36.2 | 0.333 | 0.256 |

**Context option bias tends to manifest more prominently in the deeper layers of the model**. In Figure 5, we analyze the variations in attention allocation for different types of MCQ. The model

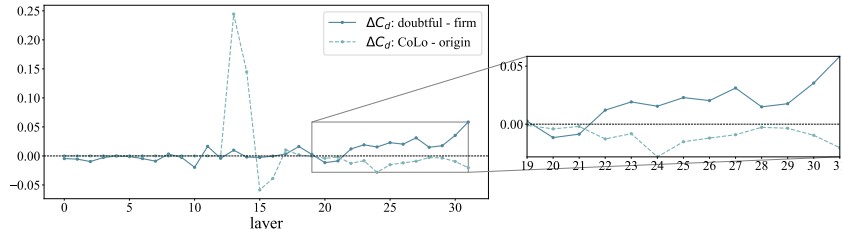

Figure 5: We utilized the `Llama2-7B` model to evaluate the impact of using CoLo on attention to demonstrations within the MMLU dataset. It was observed that the model exhibited increased attention to demonstrations in deeper layers when faced with doubtfule MCQs, thereby introducing contextual option bias. By employing CoLo, the model's attention to demonstrations could be reduced, mitigating the bias introduced in deeper layers.

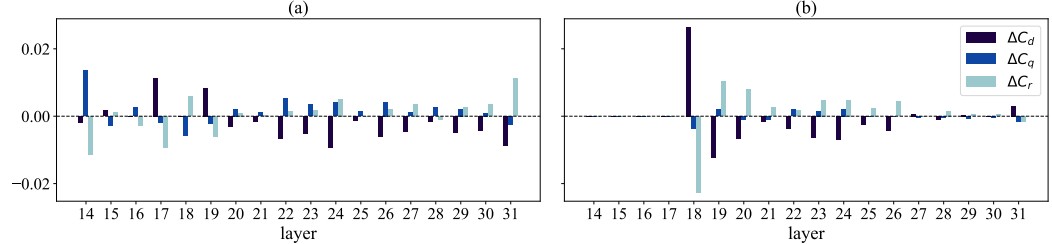

Figure 6: `Llama2-7B` model was used to evaluate the difference in attention to context demonstrations before and after the intervention. Figures (a) and (b) illustrate the head positioning in the 14th and 18th layers, respectively, following the intervention on the BBH sports understanding dataset (Suzgun et al., 2022).

tends to rely more on contextual demonstrations when handling uncertain questions, while it focuses more on the question itself for more confident ones. Based on this, we hypothesize that context-option bias primarily arises in the model's deeper layers. CoLo mitigates this bias by reducing the model's reliance on context in these deeper layers, thereby effectively decreasing context option bias.

**Deep layers reduce attention to context demonstrations following intervention**. CoLo is designed to mitigate context-option bias, which often arises from contextual information and typically manifests in the deeper layers of the model. This is supported by the observed reduction in attention to context demonstrations in the deeper layers through our method. However, at the intervention layer, attention to context demonstrations $C_d$ may increase, as shown in Figure 6(b). This suggests a trade-off between attention to examples, questions, and rationale, with the model internally adjusting to maintain a balanced approach.

## 5 CONCLUSIONS

We propose CoLo, a general localization method designed to mitigate context option bias and improve the accuracy of MCQ. Specifically, by amplifying context option bias through rearranging the order of options in demonstrations, we compare the LLMs' outputs of biased and ordinary demonstration to localizing attention heads strongly associated with context option bias. Attention scaling interventions are then applied to reduce this bias. CoLo requires only small labeled samples to localize attention heads which can be applied across different datasets. Once the attention heads requiring intervention are identified, CoLo introduces no additional inference delay. Compared to previous debiasing methods, CoLo has a natural advantage in extending to generative tasks.

While we have provided a plausible explanation for why the identified attention heads can mitigate option bias, there remains a lack of theoretical proof supporting the efficacy of this method in reducing such bias. Additionally, although the selection of attention heads shows generalizability, achieving optimal results necessitates that the small labeled sample set used for positioning be identical to the final test set, which will incur additional offline computational overhead.

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

## A    REPRODUCIBILITY STATEMENT

All experiments in the paper can be completed in $4 \times$ GeForce RTX4090(24GB). All the raw versions of evaluation data are accessible from their official repositories in Table 5.

Table 5: Evaluated open-source models

| Models | URLs |
|---|---|
| Llama2-7B | https://huggingface.co/meta-llama/Llama-2-7b |
| Llama3-8B | https://huggingface.co/meta-llama/Meta-Llama-3-8B |
| Gemma-2B | https://huggingface.co/google/gemma-2b |
| Gemma-7B | https://huggingface.co/google/gemma-7b |
| Mistral-7b-v0.1 | https://huggingface.co/mistralai/Mistral-7B-v0.1 |
| Qwen2.5-0.5B | https://huggingface.co/Qwen/Qwen2.5-0.5B |
| Qwen2.5-1.5B | https://huggingface.co/Qwen/Qwen2.5-1.5B |
| Qwen2.5-3B | https://huggingface.co/Qwen/Qwen2.5-3B |

## B    INHERENT BIAS

Position bias manifests itself as inconsistencies in model predictions when the sequence of options changes but all other elements are held constant. Token bias, on the other hand, arises when the substitution of option tokens impacts predictions. Figure 7, an example of inconsistent prediction result of `Llama2-7B` due to position bias and token bias. The correct answer A initially selected by the model may arrive at by chance due to its own biases. By swapping option positions/swapping option contents LLMs will get the wrong answer.

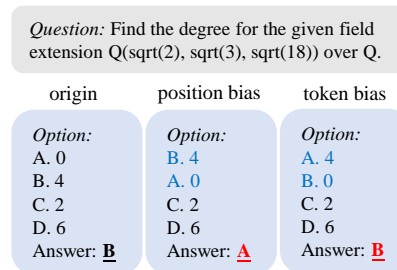

Figure 7: An example of inherent bias

## C    CONTEXT OPTION BIAS IN DEMONSTRATIONS WITH EVENLY DISTRIBUTED OPTIONS

Demonstrations with evenly distributed correct choices in a 5-shot setting yield varying effects on predictions depending on the order in which they are presented.

Table 6: With uniform distribution but different order in `Gemma-2B` result in varying predictions.

| Sequence of demonstraion | A | B | C | D |
|---|---|---|---|---|
| Random | 1094 | 4886 | 5295 | 2767 |
| A-B-C-D-A | 736 | 5833 | 5215 | 2258 |
| D-C-B-A-D | 1202 | 5849 | 5526 | 1465 |

## D    COMPARISON OF ATTENTION DISTRIBUTION

The experiments find that even with simple option swaps in demonstrations, the attention map's distribution in an LLM varies significantly as shown in Figure8.

Based on the difference in attention distribution as shown in Figure 9, we speculate that attention scaling can be used to strengthen attention to the question to make the attention distribution in the two states closer.

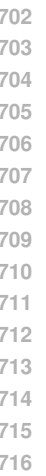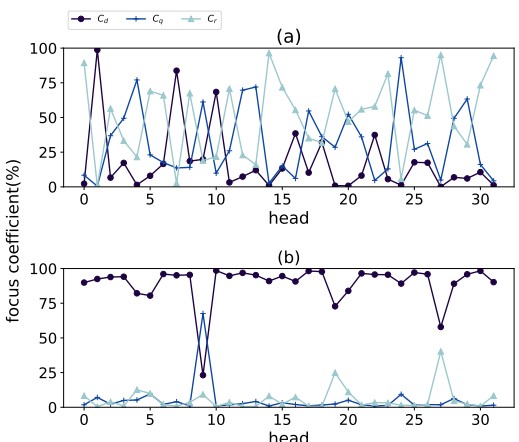

Figure 8: Utilizing the `Llama2-7B` model, we report the average value of the attention coefficients for every head in layer 14th when applied to the BBH's sports understanding dataset(Suzgun et al., 2022). (a) ordinary demonstrations (b) biased demonstrations

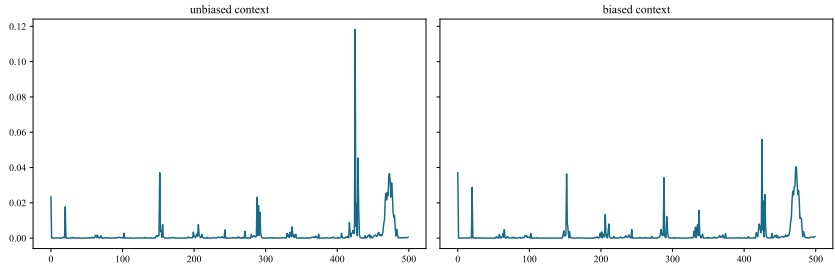

Figure 9: We used biased and ordinary demonstrations on the BBH dataset to obtain the attention distribution on head 9th and layer 14th respectively.

## E  ADDITIONAL COMPUTATIONAL OVERHEAD OF COLO

The model consists of $L$ candidate layers, with each layer containing $H$ attention heads. Given a sample size of $N$, the number of additional inferences required is:

$$L \times N \times 2 + K_l \times H \times N \times 2, \tag{10}$$

2 means that both biased and ordinary demonstration require one inference.

## F  LOCATED ATTENTION HEADS

CoLo is mainly divided into the offline head localization phase and the inference phase. The offline localization phase primarily consists of three steps:

1. Construct biased MCQ by altering the order of options.

2. Vote to select top $K_l$ layers.

3. Select top $K_h$ heads based on $E_f$.

Table 7 shows the attention heads we obtained by randomly selecting 80 samples from the MMLU data set and positioning them on different models. We use {*layer:(head list, T)*} to represent the positioned attention head, $K_l = 2$, $K_h = 4$.

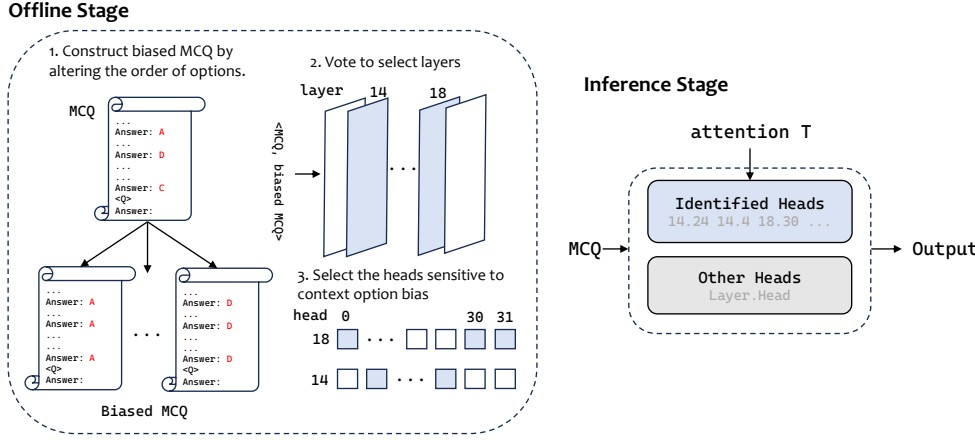

Figure 10: The overall of CoLo, encompassing offline heads identification and the inference phase.

Table 7: The attention heads obtained by CoLo of each model

| Models | Attention heads |
|--------|-----------------|
| Llama2-7B | {14: ([24, 4, 20, 31], 0.5), 18: ([30, 10, 25, 28], 0.5)} |
| Llama3-8B | {17: ([24, 25, 26, 28], 0.5), 14: ([23, 5, 4, 20], 0.5)} |
| Gemma-2B | {12: ([3, 7, 2, 1], 0.5), 14: ([0, 1, 6, 7], 0.5)} |
| Gemma-7B | {18: ([0, 8, 6, 2], 0.5), 2: ([1, 5, 3, 0], 0.5)} |
| Mistral-7B-v0.1 | {16: ([12, 14, 13, 1], 0.5), 19: ([8, 16, 9, 10], 0.5)} |
| Qwen2.5-0.5B | {15: ([13, 12, 9, 7], 0.5), 14: ([0, 3, 6, 11], 0.5) } |
| Qwen2.5-1.5B | {21: ([11, 9, 6, 8], 0.5), 18: ([3, 11, 0, 6], 0.5)} |
| Qwen2.5-3B | {27: ([1, 4, 3, 11], 0.5) 8: ([7, 12, 3, 9], 0.5)} |

## G CALCULATE SCORE BY KL DIVERGENCE

Using the same positioning strategy and sample set, modify the equation 3 to following equation 11. $\mathrm{KL}(\cdot)$ represents the calculation of KL divergence.

$$score = 1 - \mathrm{KL}(p'(y_o|d_b, q), p(y_o|d_b, q)) + p'(y_t|d_b, q) - p(y_t|d_b, q) \qquad (11)$$

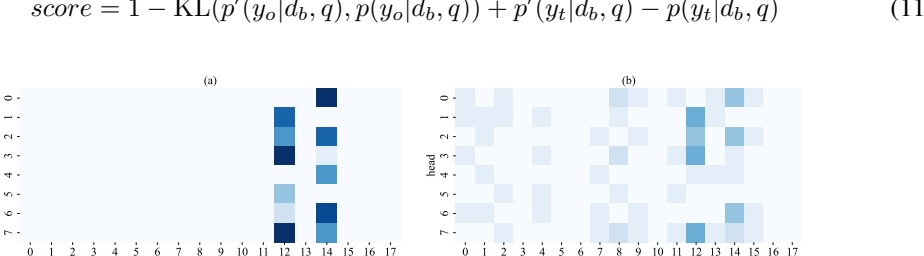

Figure 11: (a) calculate score by equation 3.(b) Modify the original formula through KL divergence.

## H COMPARISON OF CROSS-DOMAIN MMLU PERFORMANCE

Table 2 presents the performance of non-cross-domain CoLo and PriDe on the MMLU dataset. To compare the cross-domain transfer capabilities of CoLo and PriDe, we conducted cross-domain experiments on MMLU under different settings as shown in Table 8. The MMLU dataset consists of four domains: STEM, Social Science, Humanities, and Others. For both CoLo and PriDe, offline computations were performed within a single domain to prepare for debiasing. Specifically, PriDe calculates the prior probability, while CoLo identifies the relevant attention heads. Accuracy across the various domains is then obtained through cross-domain evaluation, and the average for each domain is calculated to determine the overall accuracy on MMLU.

Table 8: Cross-domain comparative experiment between CoLo and PriDe on MMLU.

| MODEL Cost MMLU-Transfer | Baseline ×1 | | CoLo ≈ ×1.5 | | PriDe(5%) ×1.15 | | Pride(40%) ×2.2 | | Pride(40%)+CoLo ×3.7 | |
|---|---|---|---|---|---|---|---|---|---|---|
| | Acc | RStd | Acc | RStd | Acc | RStd | Acc | RStd | Acc | RStd |
| Gemma-2B | 33.8 | 21.8 | 36.2 | 12.6 | 33.8 | 10.1 | 35.5 | 7.6 | **37.8** | **7.5** |
| Gemma-2B+FS | 39.9 | 14.3 | 41.7 | 10.8 | 41.6 | 6.4 | 42.4 | 5.1 | **42.9** | **4.6** |
| Gemma-7B | 61.6 | 5.2 | 62.0 | 5.5 | 61.7 | 6.7 | 62.3 | 5.5 | **62.5** | **5.1** |
| Gemma-7B+FS | 62.6 | 5.4 | 63.4 | **3.7** | 64.3 | 5.9 | 64.7 | 4.9 | **64.5** | 4.2 |
| Llama2-7B | 40.9 | 13.9 | 43.2 | **4.4** | 40.0 | 9.2 | 41.7 | 7.6 | **44.3** | 6.9 |
| Llama2-7B+FS | 45.5 | 10.7 | 46.5 | **3.8** | 45.5 | 9.0 | 46.5 | 7.4 | **47.4** | 8.0 |
| Llama3-8B | 61.4 | 13.8 | 61.9 | 6.8 | 63.4 | 5.9 | **63.7** | 4.8 | 63.1 | 4.0 |
| Llama3-8B+FS | 64.6 | 7.8 | 64.8 | 4.7 | 65.0 | 3.9 | **65.4** | 3.1 | 64.7 | **2.7** |

Table 9: Performance of CoLo on MMLU with different $K_l$ and $K_h$

| $K_l \backslash K_h$ | 1 | 2 | 4 | 6 | 8 |
|---|---|---|---|---|---|
| 1 | 40.6/12.3 | 40.9/11.8 | 41.7/10.7 | 41.1/11.3 | 41.0/10.9 |
| 2 | 41.3/11.8 | 41.4/10.7 | 41.7/**8.8** | 41.2/9.9 | 41.2/10.6 |
| 3 | 41.5/11.6 | 41.5/11.3 | 41.2/9.8 | 41.2/10.0 | 41.0/10.1 |
| 4 | 40.9/11.1 | **41.8**/9.4 | 41.2/9.8 | 40.8/10.1 | 40.9/9.7 |
| 5 | 40.9/11.8 | 41.1/10.9 | 40.9/8.0 | 41.0/11.4 | 39.4/11.7 |
| 6 | 40.8/11.2 | 41.0/10.6 | 40.8/10.8 | 40.4/11.1 | 38.4/10.7 |

# I SUPPLEMENTARY EXPERIMENTS

We explored $K_l$ and $K_h$ using Gemma-2B model in Table 9. When the number of heads is between 6 and 8, the performance remains consistent; however, as the number of heads increases further, the effectiveness of the method diminishes. Similarly, the number of intervention layers should not be excessive. Based on our findings, we recommend using 2–3 intervention layers and 6–8 heads as the most suitable configuration. Furthermore, with regard to the intervention layer, our experiments on various models have identified a pattern: the optimal intervention layer is typically located in the middle layers of the model.

In our study, we initially selected 10 samples but did not investigate or elaborate on how sample size influences the method's performance. To address this limitation, we conducted additional experiments using Gemma-2b on the MMLU benchmark, varying the number of samples. Each experiment was repeated five times, with the mean accuracy and Rstd reported. The results demonstrate that when the sample size reaches 10, the performance improvement is approaching a plateau. Based on these findings, we estimate that the minimum number of samples required is approximately 10–12 as shown in Table 10. For improved stability, we recommend using 12 samples.

Table 10: The performance scale with different number examples.

| Sample Num | 0 | 4 | 6 | 8 | 10 | 12 | 14 | 16 |
|---|---|---|---|---|---|---|---|---|
| Acc | 39.9 | 40.0 | 40.6 | 41.2 | 41.7 | 41.6 | 41.7 | 41.6 |
| Acc Var | / | 0.25 | 0.61 | 0.42 | 0.2 | 0.09 | 0.12 | 0.11 |
| RStd | 14.3 | 12.2 | 10.9 | 10.5 | 10.8 | 9.4 | 9.6 | 9.6 |
| RStd Var | / | 1.3 | 3.9 | 3.6 | 3.4 | 2.3 | 2.5 | 2.1 |

Table 11: Supplementary experiments on the Qwen2.5 series model using CoLo

| Model | Cost | Zero Shot | | Ordinary (5 shot) | | Biased (5 shot) | |
|---|---|---|---|---|---|---|---|
| | | Acc(%) | RStd(%) | Acc(%) | RStd(%) | Acc(%) | RStd(%) |
| Qwen2.5-0.5B | 1 | 46.2 | 17.2 | 47.5 | 13.7 | 47.5 | 13.7 |
| +CoLo | ×1.10 | 47.4 | 13.5 | 48.6 | 7.9 | 48.6 | 7.9 |
| Qwen2.5-1.5B | 1 | 58.8 | 8.4 | 59.4 | 3.6 | 59.4 | 3.6 |
| +CoLo | ×1.19 | 59.6 | 7.5 | 60.5 | 0.9 | 60.5 | 1.0 |
| Qwen2.5-3B | 1 | 64.2 | 8.3 | 65.6 | 3.2 | 65.2 | 3.2 |
| +CoLo | ×1.25 | 65.4 | 6.3 | 66.4 | 1.4 | 66.2 | 1.4 |

