# OpenReview forum: "Large Language Models Can Be More Robust Multiple Choice Selectors Through Attention Intervention"
_ICLR.cc/2025/Conference — Submitted to ICLR 2025_

### Official Review · Reviewer_hFgX · 2024-11-02

**Soundness:** 3
**Presentation:** 3
**Contribution:** 3
**Rating:** 6
**Confidence:** 4

**Summary:**

This paper presents an approach, termed CoLo, to address the bias in multiple-choice question (MCQ) selection that arises from context option bias in large language models (LLMs). CoLo mitigates context option bias by intervening on specific attention heads that are highly sensitive to biased in-context demonstrations. By selectively scaling attention in these heads, CoLo reduces the model's predisposition to biased options, showing improvements in MCQ accuracy across several datasets. The authors also conduct extensive experiments to validate the cross-domain robustness of their method, indicating that CoLo's effect generalizes well across various MCQ scenarios.

**Strengths:**

* I appreciate the focus on context option bias, a previously-overlooked form of bias that can impact LLM performance in MCQ tasks. The design of CoLo is also well-motivated.
* The paper provides extensive empirical evidence that shows improvements in MCQ accuracy and recall standard deviation. The authors also consider both zero-shot and few-shot scenarios and the cross-dataset setting, enhancing the generalizability of their findings.

**Weaknesses:**

I would like to see evaluation with more larger models, as the current experiments are conducted with at most 7B/8B-level models. So I am not fully convinced about CoLo's scalability. For example, how CoLo would perform with 30B/70B-level, or MoE models (these experiments may be conducted with Qwen2/Qwen2.5 models, as LLaMA-3 does not have too many variants)? Also, what the specific GPU hardware and hours will be needed for various-sized models? The lack of experiments and analysis on larger models may limit the broader impact and implications of the proposed method.

**Questions:**

1. Identifying the attention heads most associated with context option bias requires multiple rounds of sampling, which could add computational complexity in real-world applications (and higher than the PriDe baseline). While this overhead is justified within the paper’s scope, discussing potential optimizations or alternative approaches might help broaden CoLo’s applicability, especially for larger-scale implementations.
2. While the concept of CoLo is sound, the process for selecting and intervening on attention heads may seem somewhat convoluted to readers not already familiar with the technical nuances of attention heads and scaling interventions. Additional clarifications or illustrative examples could enhance the accessibility of the proposed approach.

---

> ### Author Response · Authors · 2024-11-21
>
> We are delighted by your appreciation of our paper and your recognition of the well-motivated design of CoLo.
>
> **Question**
>
> **Q1: Computational overhead of CoLo and potential optimization directions.**
>
> > Identifying the attention heads most associated with context option bias requires multiple rounds of sampling, which could add computational complexity in real-world applications (and higher than the PriDe baseline).
>
> We sincerely appreciate your thorough understanding of CoLo and your attention to the comparison of computational overhead between CoLo and PriDe.
> We would like to clarify the following two points:
> 1. CoLo operates offline, which means it does not introduce additional computational overhead during inference.
> 2. The additional computational overhead of CoLo is independent of the dataset size, whereas PriDe's overhead increases with the growth of the dataset.
>
> Therefore, we believe that CoLo also has its unique advantages in practical applications.
>
> > While this overhead is justified within the paper’s scope, discussing potential optimizations or alternative approaches might help broaden CoLo’s applicability, especially for larger-scale implementations.
>
> This is an exceptionally insightful question, as it highlights the additional computational overhead introduced by CoLo in large-scale deployment. We have detailed the computational cost associated with localization in Appendix E, which is proportional to the number of layers and heads in the model. As the model size increases, the associated computational overhead also grows. Specifically, for models scaling from Llama2-7B to Llama2-70B, the number of layers increases from 32 to 80, and the number of heads increases from 32 to 64. Consequently, the localization overhead is expected to grow by a factor of 4.5, calculated as 80/32+64/32.
> The following are some potential directions for optimization:
> - The computational overhead can be mitigated by employing a grouping strategy for layers and heads. For instance, adjacent layers and heads can be combinedcan. This approach effectively reduces the Llama2-70B to "40 layers" and "32 heads", resulting in additional computational overhead comparable to that of Llama2-7B.
> - In multi-round voting for layers, the layers with lower rankings can be eliminated in each round.
>
> **Q2: Additional clarifications or illustrative examples could enhance the accessibility of the proposed approach.**
>
> > While the concept of CoLo is sound, the process for selecting and intervening on attention heads may seem somewhat convoluted to
> readers not already familiar with the technical nuances of attention heads and scaling interventions.
>
> Thank for your valuable suggestions. We have included the workflow diagram of CoLo (Figure 10) in Appendix F. We are considering whether to incorporate it into the introduction section in future versions, as this could enhance readers' understanding of the CoLo process.

---

> > ### Comment · Reviewer_hFgX · 2024-11-30
> >
> > Thanks for your clarification. I will maintain my positive rating and lean toward acceptance.

---

### Official Review · Reviewer_G8PB · 2024-11-04

**Soundness:** 3
**Presentation:** 3
**Contribution:** 2
**Rating:** 5
**Confidence:** 2

**Summary:**

Paper proposes a new approach to identify and intervene with attention heads in order to minimize biases from in-context examples

**Strengths:**

_ Proposed approach is novel and the problem of biased in-context examples in important
- Paper is clearly written and easy to understand
- Experiments are thorough, including different models and tasks
- Results show that the approach leads to improved performance and is robust to the style of the input and the dataset

**Weaknesses:**

- Results mostly show 1-3% improvement, which is nice, but not a huge improvement
- Experiments are focused on multiple choice. It would be great to show that the proposed method is able to generalize to more open ended question answering tasks, which are more representative of realistic tasks that LLMs are used for
- The proposed method requires both original and biased versions of in-context examples. However, in more open ended question, LLMs may still suffer from biased in-context examples, but it may not be clear what the bias is. Is there a way to not required both biased and unbiased versions of in-context examples in the method?

**Questions:**

See weaknesses

---

> ### Author Response · Authors · 2024-11-20
>
> **Glossary**
>
> RStd: Recall standard deviation (RStd), an indicator for measuring model's option bias, measuring the balance of recall rates across different option IDs. A smaller RStd reflects a lower level of option bias in the model.
>
> ---
> Thank you for your valuable and constructive review. We are pleased that you think our proposed approach novel.
>
> > W1：Results mostly show 1-3% improvement, which is nice, but not a huge improvement
>
> We sincerely appreciate the reviewer’s recognition of our work. Our primary goal is not only to enhance accuracy but also to mitigate the impact of option bias on the model. Allow us to elaborate on a few key points:
> - Beyond the observed accuracy gains, the reduction of another metric, RStd, is particularly noteworthy, as it objectively reflects the model's ability to alleviate option bias.
> - Similar study PriDe[1] has also achieved only modest accuracy improvements, our approach surpasses PriDe in reducing RStd and improving accuracy, particularly in cross-dataset generalization. Furthermore, we observe that combining our method with PriDe yields even better performance.
> - CoLo consistently demonstrats accuracy improvements across different models, indicating it's broad applicability.
>
> > W2：Experiments are focused on multiple choice. It would be great to show that the proposed method is able to generalize to more open ended question answering tasks, which are more representative of realistic tasks that LLMs are used for.
>
> We believe this is an excellent and bold suggestion, well worth exploring further. This is why we are committed to employing inference-based interventions rather than relying on prior-based debiasing methods, as seen in previous work such as PriDe[1]. However, extending this approach to open-ended generation would require substantial additional work, which lies beyond the scope of this paper. MCQ paradigm holds significant importance, particularly in model evaluation. Consequently, our method currently focuses on demonstrating its effectiveness specifically in mitigating option bias.
>
> > W3: The proposed method requires both original and biased versions of in-context examples. However, in more open ended question, LLMs may still suffer from biased in-context examples, but it may not be clear what the bias is. Is there a way to not required both biased and unbiased versions of in-context examples in the method?
>
>
> As previously mentioned, our method is currently effective solely in mitigating option bias. Addressing other forms of implicit bias in open-domain generation, such as those related to sentiment or competence,  remains an open challenge and a promising direction for future research. We believe this represents a valuable and worthwhile avenue for further exploration.
>
> We hope to have been able to address all the reviewers’ concerns, are happy to answer any follow-up questions they might have, and are looking forward to their reply.
>
> **References**
>
> [1] Zheng C, Zhou H, Meng F, et al. Large language models are not robust multiple choice selectors[C]//The Twelfth International Conference on Learning Representations. 2023.

---

> ### Comment · Reviewer_G8PB · 2024-11-25
>
> Thanks to the authors for the reply.
>
> I would like to maintain my score, as I believe reducing options bias in multiple choice questions is a very narrow problem setting which may have relatively low relevance in practical applications of LLMs.
>
> It would be great to expand the scope of the work to include a wider range of biases in tasks which allow for more open-ended model responses.

---

> > ### Author Response · Authors · 2024-12-02
> >
> > Thanks for your response.
> >
> > We fully agree with the suggestion to expand the scope to include a broader range of biases. However, option bias remains a crucial issue, and a significant body of research has focused on this topic [1,2,3,4].
> >
> > Currently, CoLo has limitation that it need to swap the order of options. Inspired by the feedback from Reviewer DxNU, we have identified that **using a prompt-based method can also effectively generate sample pairs**. We have conducted preliminary experiments in this direction. This simplified approach is highly conducive to extending the study to a wider variety of biases, particularly in open-domain settings. Due to time constraints, we were unable to conduct more detailed experiments during the rebuttal period. We are confident that our approach has significant potential for scaling to incorporate additional types of biases.
> >
> > [1] Pezeshkpour P, Hruschka E. Large language models sensitivity to the order of options in multiple-choice questions
> >
> > [2] Wang H, Zhao S, Qiang Z, et al. Beyond the answers: Reviewing the rationality of multiple choice question answering for the evaluation of large language models
> >
> > [3] Balepur N, Ravichander A, Rudinger R. Artifacts or Abduction: How Do LLMs Answer Multiple-Choice Questions Without the Question?
> >
> > [4] Choi H K, Xu W, Xue C, et al. Mitigating selection bias with node pruning and auxiliary options

---

### Official Review · Reviewer_Cigt · 2024-11-04

**Soundness:** 2
**Presentation:** 2
**Contribution:** 2
**Rating:** 5
**Confidence:** 4

**Summary:**

The paper proposes  a method  aiming at mitigating context option bias in multiple-choice questions taks, by comparing the LLMs’ outputs for biased and ordinary prompts in order to localize attention heads associated with the context option bias. Attention scaling interventions is then employed. Experiments on several dataset are described in order to validate the method, although the standard deviations are not reported.

**Strengths:**

-Paper addresses the weaknesses in large language models related with content bias in multiple-choice questions task (MCQ).

-The paper claims that the proposed method is demonstrated empirically to reduce the content bias in MCQ task on MMLU, although the standard deviations in the experiments are not  reported.

**Weaknesses:**

1)The paper provides no theoretical explanation for why only specific attention heads contribute to bias or why scaling the specific attention heads mitigates the bias effectively. This lack of theoretical grounding may affect the reproducibility and understanding of the method, making it difficult to predict its behavior across other models.

2)The standard deviations are not reported in the experiments. Especially in the view of the small differences in several important experiments, this makes impossible to evaluate the method empirically.

2)Narrow focus on attention scaling as the intervention technique. The paper uses exclusively the attention scaling to address bias without exploring other possible intervention methods, such as layer-based adjustments or targeted fine-tuning. This narrow approach may miss alternative solutions that could yield more robust or efficient debiasing effects.

3)Recent large language models, such as  for example Llama 3.2, Mistral, Qwen 2 were not used in experiments, do they suffer from similar biases? Is the proposed method effective in these cases?

4)Reliance on artificially biased prompts. The study constructs heavily biased demonstrations to amplify context option bias and identify attention heads associated with this bias. This contrived setup may not accurately reflect the subtler biases encountered in real-world applications, potentially limiting the generalizability of the findings. It is not clear why reducing the bias in the artificially biased scenario will lead to increased performance in real-world applications.

5)Sensitivity to hyperparameter choice. The method has hyperparameters, such as the scaling factor T for attention adjustments, but the paper  analysis of how sensitive the model's performance is to these choices is limited. Without detailed guidance for these hyperparameters, reproducing the results and applying the method to new datasets or models would require extensive tuning, which can be  resource-intensive.

6)Limited evaluation of cross-domain generalizability.  The evaluation of cross-dataset robustness is limited to certain datasets and tasks, where the reported small change is impossible to validate in the absence of  standard deviations reported. A more comprehensive assessment across a broader range of domains would be necessary to confirm the method’s generalizability, especially given the diverse biases in data sources.

7)The readability and clarity of the paper can be improved. The paper often employs long sentences which can obscure the main points. Sentences like 'To better observe the LLMs’ behavior when affected by the context option bias, we deliberately use demonstrations with obvious context option bias for MCQ to amplify the effect' are long and somewhat repetitive. Table 2 is discussed briefly without highlighting the most important findings or explaining how they support the claims. Redundant phrases and wordiness occasionally affect readability. For example, phrases like 'To better excavate the reasons behind this problem and resolve it...' could be made more concise. 'One of effective methods' requires 'the' article here; '...significantly degrade LLMs’ performance' as  'context option bias' is singular, so 'degrade' should be 'degrades.'  Vague pronoun references, in a few instances, pronouns (like 'it' or 'this') refer ambiguously to previous sentences, making the meaning less clear. 'The experiments find that... This inspires us to test...' — here, "This" could refer to several prior ideas, so specifying 'This result' or 'This observation' would improve clarity.

8)The captions under Figure 8 are somewhat misleading, how to read the frequency of the localized field from picture (c)?

**Questions:**

Did you experiment with other recent llms, such as Llama 3.2, Mistral, Qwen 2, do they suffer from similar biases? Is the proposed method effective in these cases?

Why reducing the bias in such artificially biased scenario will lead to better performance in real-world applications?

Did you try other options of reducing the bias, like targeted fine-tuning?

Why the standard deviations in the experiments were not reported?

How to tune the hyperparameters of the method such as T? the performance seems to depend heavily on its value (see table 1)

---

> ### Author Response · Authors · 2024-11-20
>
> **Glossary**
>
> RStd: Recall standard deviation (RStd), an indicator for measuring model's option bias, measuring the balance of recall rates across different option IDs. A smaller RStd reflects a lower level of option bias in the model.
>
> ---
>
> Thank you for your insightful and constructive review. To address your primary concern regarding the reporting of standard deviations, we have conducted additional experiments and performed a more extensive evaluation of the parameters used in CoLo.
>
> **Q1: Did you experiment with other recent llms, such as Llama 3.2, Mistral, Qwen 2, do they suffer from similar biases? Is the proposed method effective in these cases?**
> > W3: Recent large language models, such as for example Llama 3.2, Mistral, Qwen 2 were not used in experiments, do they suffer from similar biases? Is the proposed method effective in these cases?
>
> We conducted experiments on Qwen2.5 and report the debiasing effect of CoLo on this model. We observed that Qwen2.5 appears to be less affected by context option bias, as the accuracy and Rstd under both the Ordinary 5-shot and Biased 5-shot settings are nearly identical. Nevertheless, CoLo still demonstrates performance improvements. We hypothesize that this could be because CoLo not only mitigates context option bias but may also alleviate inherent option bias to some extent. However, this hypothesis was not explicitly stated in our work due to the lack of experimental validation. We present this result in Appendix I.
>
> |Model| 0-shot (Acc/RStd)| Ordinary 5-shot (Acc/RStd) | Biased 5-shot (Acc/RStd)|
> |-|:-:|:-:|:-:|
> |Qwen2.5-0.5B|  46.2/17.2 | 47.5/13.7 | 47.5/13.7 |
> |Qwen2.5-0.5B + CoLo|  **47.7**/**13.5** | **48.6**/**7.9** | **48.6**/**7.9** |
> |Qwen2.5-1.5B|  58.8/8.4 | 59.4/3.6 | 59.4/3.6 |
> |Qwen2.5-1.5B + CoLo| **59.6**/**7.5** | **60.5**/**0.9** | **60.5**/**1.0** |
> |Qwen2.5-3B| 64.2/8.3 | 65.6/3.2 | 65.2/3.2 |
> |Qwen2.5-3B + CoLo|  **65.4**/**6.3** | **66.4**/**1.4** | **66.2**/**1.4** |
>
> **Q2: Why reducing the bias in such artificially biased scenario will lead to better performance in real-world applications?**
>
> Our experiments reveal that implicit biases can exist even in seemingly unbiased scenarios. CoLo is not limited to artificially constructed option-biased scenarios, it is equally applicable in general few-shot settings and can help mitigate the introduction of implicit option biases.
>
> **Q3: Did you try other options of reducing the bias, like targeted fine-tuning?**
> > W2: Narrow focus on attention scaling as the intervention technique.
>
> We sincerely thank the reviewer for their insightful questions. Our work does not employ targeted fine-tuning for the following reasons:
> 1. Inspired by prior research[3], which suggests that different attention heads perform distinct functions, our objective is to directly identify attention heads associated with option biases within the model. This approach holds greater significance for enhancing model interpretability.
> 2. We believe that our method is complementary to fine-tuning and operates in an orthogonal manner.
> 3. Compared to fine-tuning, our approach requires only a minimal number of additional samples.
>
> **Q4: Why the standard deviations in the experiments were not reported?**
>
> During the head localization process in CoLo, additional samples were required. These samples were drawn from the development set, which was also used to construct the few-shot prompt. These samples were randomly selected but remained fixed in subsequent experiments. The identified heads are documented in the appendix. We appreciate the feedback and recognize the importance of conducting multiple replicates to evaluate the algorithm's stability. For instance, using Gemma-2B as a case study, we plan to explore different sample sizes, perform repeated experiments, and report the mean and standard deviation of the accuracy improvements.
>
> |Sample Num|0|4|6|8|10|12|14|
> |-|:-:|:-:|:-:|:-:|:-:|:-:|:-:|
> |Acc|39.9|40.0(±0.25)|40.6(±0.61)|41.2(±0.42)|**41.7**(±0.20)|41.6(**±0.09**)|41.7(±0.12)|
> |RStd|14.3|12.2(±1.3)|10.9(±3.9)|10.5(±3.6)|10.8(±3.4)|**9.4**(**±2.3**)|9.6(±2.5)|
>
>
> **Q5: How to tune the hyperparameters of the method such as T? the performance seems to depend heavily on its value (see table 1)?**
>
> We conducted ablation experiments with different values of  T  on Gemma 2B, as shown in Figure 4(c), and found that T within the range [0.3, 0.6] yields significant improvements, with the best performance observed at T = 0.4.
>
> |T|0.1|0.2|0.3|0.4|0.5|0.6|0.7|0.8|0.9|1.0|1.1|1.2|1.3|1.4|
> |-|:-:|:-:|:-:|:-:|:-:|:-:|:-:|:-:|:-:|:-:|:-:|:-:|:-:|:-:|
> |Baised $\delta$Acc(%)| 2.03|2.34|2.61|**2.72**|2.59|2.30|1.75|1.21|0.59|0.0|-0.71|-1.09|-1.65|-2.12|
> |Ordinary $\delta$Acc(%) |1.62|1.73|1.76|**1.82**|1.74|1.41|1.10|1.04|0.47|0.0|-0.93|-1.45|-2.22|-2.84|
>
> In  our study, although we did not identify the optimal parameters, we applied the same settings of T=0.5, across different models, all of which demonstrated accuracy improvements.

---

> ### Author Response · Authors · 2024-11-20
>
> **Weaknesses**
>
> > W1: The paper provides no theoretical explanation for why only specific attention heads contribute to bias or why scaling the specific attention heads mitigates the bias effectively.
>
> We acknowledge that CoLo lacks a comprehensive theoretical explanation and instead relies on experimental evidence, as emphasized in the Conclusion section. This reliance arises from the inherent challenges in conducting theoretical analyses of the behavior of large models—a difficulty that has led many studies, including ours, to adopt an experimentally driven approach [1,2]. Our research is grounded in the premise that different attention heads perform distinct functional roles [3], and we posit that interventions specific attention heads can mitigate option bias.
>
> > W5: Sensitivity to hyperparameter choice. Without detailed guidance for these hyperparameters, reproducing the results and applying the method to new datasets or models would require extensive tuning, which can be resource-intensive.
>
> Thank you for your valuable suggestions regarding our experiments. In response, we conducted additional experiments to investigate the impact of the sample number, $K_l$, and $K_h$​ on CoLo. The results provide useful insights into parameter selection. Based on our analysis, we recommend setting the sample number to 12, $K_l$ and $K_h$​ to [2, 4] or [4, 2], and T=0.5.  In  our study, although we did not identify the optimal parameters, we applied the same settings of T=0.5, $K_l$= 2, and $K_h$=4 across different models, all of which demonstrated accuracy improvements. This suggests that the approach possesses a certain degree of generalizability.
>
> > W6: Limited evaluation of cross-domain generalizability.
>
> We utilize the heads identified through localization on the MMLU dataset to evaluate their performance on other multiple-choice datasets (CMMLU, CEVAL, and AGIEVAL) in Table 3, thereby demonstrating the general applicability of these localized heads. We did not sufficiently emphasize the Table 3. To address this, we have now highlighted "using the attention heads obtained from MMLU" in bold in the caption.
>
> > W7: The readability and clarity of the paper can be improved.
>
> Thank you for the careful reading and correcting the errors. We revised the original sentence, "To better observe the LLMs' behavior when affected by the context option bias, we deliberately use demonstrations with obvious context option bias for MCQ to amplify the effect." to "We deliberately design demonstrations with an obvious context option bias in MCQs to amplify the effect, allowing for clearer observation of differences in LLMs' behavior."
>
> We expanded the title of Table 2. The identified grammatical errors have been corrected.
>
> > W8: The captions under Figure 8 are somewhat misleading, how to read the frequency of the localized field from picture (c)?
>
> Thank you for pointing this out. You may be referring to Figure 4(c). Upon review, we noticed an issue with the order of the labels (a), (b), and (c) in the figure. We have identified and corrected this mistake.
>
>
> **References**
>
> [1] Conmy A, Mavor-Parker A, Lynch A, et al. Towards automated circuit discovery for mechanistic interpretability[J]. Advances in Neural Information Processing Systems, 2023, 36: 16318-16352.
>
> [2] Goldowsky-Dill N, MacLeod C, Sato L, et al. Localizing model behavior with path patching[J]. arXiv preprint arXiv:2304.05969, 2023
>
> [3] Fred Zhang and Neel Nanda. Towards best practices of activation patching in language models:
> Metrics and methods, 2024.

---

> ### Author Response · Authors · 2024-11-30
>
> Dear Reviewer Cigt,
>
> We sincerely appreciate your time and effort in reviewing our manuscript and offering valuable suggestions. We provided detailed clarifications in response to your questions a few days ago. If you have any additional feedback, concerns, or questions regarding our response, we would greatly appreciate hearing from you and welcome further discussion.

---

> > ### Comment · Reviewer_Cigt · 2024-12-01
> >
> > I have read the rebuttal and the discussions with other reviewers. Some of the issues were addressed, I raise the score accordingly, however some issues remain, including somewhat weak reporting of the algorithm stochastic stability, and   somewhat incremental nature of the results. I agree with Reviewer G6PB that the problematic of the studied bias is somewhat narrow and the paper would benefit from studying applicability of the method to other known biases.

---

> ### Author Response · Authors · 2024-12-02
>
> We glad that our response has addressed some of your concerns, and we sincerely appreciate your decision to raise the score. We fully agree with the idea of expanding to include a wider range of biases. However, option bias is also a crucial issue, and many studies have focused on this topic[1,2,3,4].
>
> Currently, CoLo has certain limitations, such as the need to swap the order of options.  Inspired by the reviewer DxNU's feedback, we have discovered that **using a prompt-based method can also effectively generate sample pairs**, and we have conducted some preliminary experiments in this direction. This simplified construction method is highly conducive to extending the study to a broader range of biases, particularly in open-domain settings. Due to time constraints, it was not feasible to conduct detailed experiments during the rebuttal period. However, we are confident that our approach has significant potential for scaling to more types of biases.
>
> [1] Pezeshkpour P, Hruschka E. Large language models sensitivity to the order of options in multiple-choice questions
>
> [2] Wang H, Zhao S, Qiang Z, et al. Beyond the answers: Reviewing the rationality of multiple choice question answering for the evaluation of large language models
>
> [3] Balepur N, Ravichander A, Rudinger R. Artifacts or Abduction: How Do LLMs Answer Multiple-Choice Questions Without the Question?
>
> [4] Choi H K, Xu W, Xue C, et al. Mitigating selection bias with node pruning and auxiliary options

---

### Official Review · Reviewer_DxNU · 2024-11-06

**Soundness:** 3
**Presentation:** 3
**Contribution:** 2
**Rating:** 6
**Confidence:** 4

**Summary:**

This paper proposes a method to interpret which attention cells are responsible for “context option bias” — a form of bias that occurs when an LLM inclines to generating one option more than others in an in-context learning scenario. The authors propose a difference metric that measures the change in attention values when biased or ordinary examples are used with in-context learning. This metric is also used to detect top-layers that are the most important for context option bias. They also propose a solution to alleviate the impact of bias: scaling attention values of the detected cells using a custom temperature. Experimental results suggest that the proposed method improves MMLU performance and detected attention heads transfer to other domains such as CEVAL. The detected heads are robust to different number of choice or demonstrations but sensitive to the particular biased option.

**Strengths:**

Update: Authors provided detailed explanations with many additional experiments which addressed my concerns. I increased my score accordingly.

1. The custom temperature per attention head is a simple but affective approach for detecting context option bias.

2. Experimental results indicate success with better detection and improved performance.

3. Transferability to other tasks indicate that these heads are feature of the model and pre-training rather than downstream domain. This is an interesting finding.

**Weaknesses:**

There are a few concerns that I detailed below.

1. I think a better motivation on why context option bias for MCQ is a practical problem is needed. Given that options in in-context examples can be shuffled freely, this needs a better motivation.

2. What is the minimal number of examples for successful detection of bias? More specifically, how does the performance scale with more examples?

3. What if you add a custom prompt message that explains that “there are multiple options, the answer can be any one of them, and no particular option such as A or B is more likely.” Would this be an affective strategy to steer the attention heads?

4. How does the problem change with increasing model size? Are larger models more or less susceptible?

5. Equation 2 should be $y=argmax$.

6. How did you choose $K_l$ and $K_h$? How does the performance change as $K_l$ and $K_h$ increases?

7. Previous work has used different temperatures for different domains (see [1] for one such work) that are not necessarily 1.0. Can you run a search over different temperatures for the baseline and report the best performing model?

8. Can you explain why the performance of Llama-3 and Gemma models are very different from official numbers from respective papers?

9. Can you explain what do you mean by confidence in Section 4.5?

10. Caption of the Figure-5 is not grammatically sound. Please rewrite.

[1] Large Language Monkeys: Scaling Inference Compute with Repeated Sampling. Bradley Brown and Jordan Juravsky and Ryan Ehrlich and Ronald Clark and Quoc V. Le and Christopher Ré and Azalia Mirhoseini.

**Questions:**

1. Why does context option bias for MCQ a practical problem?

2.  How does the performance scale with more examples?

3. What happens if you use a custom prompt message?

4. What is the impact of model size on the problem?

5. How does the performance change w.r.t. different $K_l$ and $K_h$? How did you choose these parameters?

6. Can you improve the baseline by finding the best temperature?

7. Why does the performance of Llama-3 and Gemma different from official numbers?

---

> ### Author Response · Authors · 2024-11-20
>
> **Glossary**
>
> RStd:  Recall standard deviation (RStd), an indicator for measuring model's option bias, measuring the balance of recall rates across different option IDs. A smaller RStd reflects a lower level of option bias in the model.
>
> ---
> Thank you for your very helpful review, as well as your recognition of this as an interesting finding.  To address your main concerns, we will conduct additional experiments.
>
> **Q1: Why does context option bias for MCQ a practical problem?**
> > W1: Given that options in in-context examples can be shuffled freely, this needs a better motivation.
>
> We sincerely appreciate the valuable feedback and thoughtful insights. After careful consideration, we believe that the issue cannot be fully addressed by option shuffling alone. In the introduction of paragraph 3,  we discussed **some few shots may appear unbiased, but in reality they still have option bias**. We validated this through option shuffling experiments. For example, in the sequence "A-B-C-D-A," predictions for option A are reduced, while in "D-C-B-A-D," predictions for option D are reduced. There may be additional examples that were not explicitly mentioned. Therefore, we conclude that context option bias is highly prevalent.
>
> **Q2: How does the performance scale with more examples?**
>
> This question has been extremely helpful to us. In our study, we initially selected 10 samples but did not investigate or elaborate on how sample size influences the method's performance. To address this limitation, we conducted additional experiments using Gemma-2b on the MMLU benchmark, varying the number of samples. Each experiment was repeated 5 times, with the mean accuracy, Rstd and their standard deviations reported. The results demonstrate that when the sample size reaches 10, the performance improvement is approaching a plateau. Based on these findings, we estimate that the minimum number of samples required is approximately 10–12. Increasing the sample size further enhances stability, as evidenced by a reduction in variance. For improved stability, we recommend using 12 samples.
>
> We have incorporated these results into Appendix I. Once again, we are grateful for the opportunity to refine our analysis through this constructive feedback.
>
> |Sample num| 0 | 4 | 6 | 8 | 10 | 12 | 14 |
> |-|:-:|:-:|:-:|:-:|:-:|:-:|:-:|
> | **Acc** |  39.9 | 40.0(±0.25) | 40.6(±0.61) | 41.2(±0.42) | **41.7**(±0.20) | 41.6(**±0.09**) | 41.7(±0.12) |
> | **RStd** |  14.3 | 12.2(±1.3) | 10.9(±3.9) | 10.5(±3.6) | 10.8(±3.4) | **9.4**(**±2.3**) | 9.6(±2.5) |
>
> **Q3: What happens if you use a custom prompt message?**
>
> This is an excellent suggestion, and we had considered this issue initially. However, our investigation revealed that this issue has already been discussed in prior work[1]. It was suggested to prepend the prompt with the statement, 'The provided options have been randomly shuffled, so it is essential to consider them fairly and without bias.'  and observed that GPT-3.5-turbo didn't result in option bias reduction. Influenced by the conclusions of prior work, we didn't conduct this experiment in our paper. This time we utilized a custom prompt："The answer can be any one of A/B/C/D, and no particular option is more likely." to conduct experiments on the Gemma-2B model. This is an interesting experiment, and we observed that while there was some mitigation of bias in the 0-shot setting, the performance in the 5-shot setting was suboptimal.
>
> | | 0-shot (Acc/RStd) | Ordinary 5-shot (Acc/RStd) | Biased 5-shot (Acc/RStd) |
> |-|:-:|:-:|:-:|
> |Gemma-2B| 33.8/21.8 | 39.9/14.3| 38.2/16.1|
> |Gemma-2B + Custom prompt| 34.2/14.4 | 37.7/14.8| 37.6/14.8|
> |Gemma-2B + CoLo| 36.2/12.6| **41.7**/**10.8**| **40.5**/**7.0**|
> |Gemma-2B + CoLo + Custom prompt| **36.8**/**10.6**| 39.7/7.0| 39.7/7.0|
>
> **Q4: What is the impact of model size on the problem?**
> > W4: How does the problem change with increasing model size? Are larger models more or less susceptible?
>
> As the model size increases, the degree of option bias decreases. For instance, when Gemma-2b is scaled up to Gemma-7b, the RStd decreases from 14.3 to 5.4 in the 5-shot setting and from 16.1 to 10.6 in the biased 5-shot setting. Supplementary experiments yielded similar results for Qwen2.5, with model scales ranging from 0.5B to 1.5B and 3B. Notably, the use of CoLo further enhances performance beyond these improvements.
>
> |Model| 0-shot (Acc/RStd)| Ordinary 5-shot (Acc/RStd) | Biased 5-shot (Acc/RStd)|
> |-|:-:|:-:|:-:|
> |Qwen2.5-0.5B|  46.2/17.2 | 47.5/13.7 | 47.5/13.7 |
> |Qwen2.5-0.5B + CoLo|  **47.7**/**13.5** | **48.6**/**7.9** | **48.6**/**7.9** |
> |Qwen2.5-1.5B|  58.8/8.4 | 59.4/3.6 | 59.4/3.6 |
> |Qwen2.5-1.5B + CoLo| **59.6**/**7.5** | **60.5**/**0.9** | **60.5**/**1.0** |
> |Qwen2.5-3B| 64.2/8.3 | 65.6/3.2 | 65.2/3.2 |
> |Qwen2.5-3B + CoLo|  **65.4**/**6.3** | **66.4**/**1.4** | **66.2**/**1.4** |

---

> ### Author Response · Authors · 2024-11-20
>
> Additionally,  we observed that for Qwen2.5, the model appears to be less affected by context option bias, as the accuracy and Rstd under both the Ordinary 5-shot and Biased 5-shot settings are nearly identical. Nevertheless, CoLo still demonstrates performance improvements. We hypothesize that this could be because CoLo not only mitigates context option bias but may also alleviate inherent option bias to some extent. However, this hypothesis was not explicitly stated in our work due to the lack of experimental validation.
>
> **Q5: How does the performance change w.r.t. different $K_l$ and $K_h$? How did you choose these parameters?**
> > W6: How did you choose $K_l$ and $K_h$? How does the performance change as $K_l$ and $K_h$ increases?
>
> Thank you for your helpful comments, this experiment deserves to be added to the article. It is worth noting that this experiment provides valuable insights for pinpointing the appropriate heads. When the number of heads is between 6 and 8, the performance remains consistent. However, as the number of heads increases further, the effectiveness of the method diminishes. Based on our findings, we recommend using 2–3 intervention layers and 6–8 heads in total as the most suitable configuration.
> | $K_l$\\\\$K_h$| 1 | 2 | 4 | 6 | 8 |
> |-|:-:|:-:|:-:|:-:|:-:|
> |**1**|40.6/12.3| 40.9/11.8 | 41.7/10.7| 41.1/11.3| 41.0/10.9 |
> |**2**|41.3/11.8| 41.4/10.7 | 41.7/**8.8**  | 41.2/9.9 | 41.2/10.6|
> |**3**|41.5/11.6| 41.5/11.3 | 41.2/8.8  | 41.2/9.9 | 41.0/10.1|
> |**4**|40.9/11.1| **41.8**/9.4 |41.2/8.8 |40.8/10.1|40.9/9.7|
> |**5**|40.9/11.8|41.1/10.9|40.9/8.0|41.0/11.4|39.4/11.7|
> |**6**|40.8/11.2|41.0/10.6|40.8/10.8|40.4/11.1|38.4/10.7|
>
> **Q6: Can you improve the baseline by finding the best temperature?**
> > W7: Previous work has used different temperatures for different domains (see [2] for one such work) that are not necessarily 1.0. Can you run a search over different temperatures for the baseline and report the best performing model?
>
> We reviewed reference [2] as suggested by the reviewer and believe there may have been a misunderstanding regarding the concept of temperature. Different temperatures were employed for sampling across various domains[2], which is distinct from the attention scaling temperature used in our experiments. In the evaluation of MCQs, the widely accepted approach is selected as the prediction based on a greedy strategy. Due to the order-preserving property of softmax, varying the temperature coefficients doesn't affect the final result. However,  the reviewer's comments inspired us to recognize the significance of incorporating global attention scaling temperature as a comparison, rather than limiting the intervention to specific heads! Accordingly, we conducted experiments using the Gemma-2b model and reported the results. This further reinforces our rationale for intervening in only a subset of heads rather than applying changes to all of them.
>
> |T|0.2|0.4|0.6|0.8|0.9|1.0|1.1|1.2|1.4|1.6|1.8|
> |-|:-:|:-:|:-:|:-:|:-:|:-:|:-:|:-:|:-:|:-:|:-:|
> |Acc|24.4|28.4|36.8|39.4|39.8|**39.9**|39.2|37.9|35.0|29.7|25.8|
> |RStd|12.7|9.5|14.4|13.1|**13.5**|14.3|15.6|17.6|19.7|16.0|13.5|
>
> **Q7: Why does the performance of Llama-3 and Gemma different from official numbers?**
>
> The official report only provides the dataset metrics but does not include the evaluation methods. As a result, making it challenging for us to fully reproduce the reported figures. These discrepancies may stem from differences in the instruction format and the few-shot prompts used with the model. To address this, we utilized another widely recognized evaluation tool [3] to measure the accuracy improvement before and after applying CoLo in Table 3. Furthermore, to enable a meaningful comparison with prior approaches, we used the PriDe benchmark, as shown in Table 2.
>
> > W5&W10: Equation 2 should be corrected and Caption of the Figure-5 is not grammatically sound.
>
> Thanks for the correction, we have revised the manuscript.
>
> > W9: Can you explain what do you mean by confidence in Section 4.5?
>
> Thank you for pointing this out. We define confidence as the normalized probability of the selected label among all option labels, representing the model's degree of certainty in its choice. A detailed explanation of this concept has been added in Section 4.5.
>
> We hope that these clarifications, together with the analyses of new experiments, offer a more comprehensive understanding of our work. We sincerely appreciate your valuable feedback once again.
>
> **Reference**
>
> [1] Large language models are not robust multiple choice selectors. Zheng C, Zhou H, Meng F, et al. The Twelfth International Conference on Learning Representations. 2023.
>
> [2] Large Language Monkeys: Scaling Inference Compute with Repeated Sampling. Bradley Brown and Jordan Juravsky and Ryan Ehrlich and Ronald Clark and Quoc V. Le and Christopher Ré and Azalia Mirhoseini.
>
> [3] A framework for few-shot language model evaluation.

---

> ### Author Response · Authors · 2024-11-26
>
> >W3: What if you add a custom prompt message that explains that “there are multiple options, the answer can be any one of them, and no particular option such as A or B is more likely.” Would this be an affective strategy to steer the attention heads?
>
> This insight suggests that when constructing biased samples, it is not necessary to manipulate the order of options in few-shot. Instead, a prompt-based, zero-shot approach can be employed. **By including a phrase like "The answer can be any one of A/B/C/D, and option A is more likely" within the prompt**, it is possible to introduce biases toward different options (A/B/C/D). The remaining steps follow the same procedure as the original method. Through multiple experiments and reporting of the average results, both approaches identified the same layer heads: 12.3, 12.7, 12.1, 14.0, 14.1, and 14.6. This is an exciting outcome, as the prompt-based method significantly reduces the complexity of bias construction while enhancing its generalizability.
>
> |Model|Zero shot|Ordinary 5-shot|Biased 5-shot|
> |-|:-:|:-:|:-:|
> |Gemma 2B|33.8/21.8|39.9/14.3|38.2/16.1|
> |Gemma 2B +CoLo(option order)|36.2/12.6|41.7/10.8|40.5/7.0|
> |Gemma 2B +CoLo(custom prompt)|36.0/13.8|41.4/10.9|40.3/7.0|

---

> ### Author Response · Authors · 2024-11-30
>
> Dear Reviewer DxNU,
>
> We sincerely appreciate your time and effort in reviewing our manuscript and offering valuable suggestions. We provided detailed clarifications in response to your questions a few days ago. If you have any additional feedback, concerns, or questions regarding our response, we would greatly appreciate hearing from you and welcome further discussion.

---

> ### Author Response · Authors · 2024-12-03
>
> Dear Reviewer DxNU,
>
> We sincerely appreciate your time and effort in reviewing our manuscript and offering valuable suggestions. We have addressed your comments and conducted additional experiments. As the Author-Reviewer discussion period is within the final 12 hours, we would be grateful if you could confirm whether our responses have resolved your concerns.
>
> Best regards,
>
> Authors

---

### Author Response · Authors · 2024-12-03
**Summary of Rebuttal (Many Thanks to All Reviewers and AC)**

Dear Reviewers, Area Chairs, and Program Chairs,

We sincerely thank all four reviewers for their constructive comments and insightful auestions, which helped us refine our work.

*We would like to express our sincere gratitude to the reviewers for their appreciation of our work*.

**[Impactful and well motivated]**

- **Reviewer hFgX**: "The design of CoLo is also well-motivated."
- **Reviewer G8PB**: "The approach leads to improved performance and is robust to the style of the input and the dataset."
- **Reviewer DxNU**: "Transferability to other tasks indicate that these heads are feature of the model. This is an interesting finding."

**[Experiment Analysis]**

- **Reviewer hFgX**: "The paper provides extensive empirical evidence."
- **Reviewer G8PB**: "Experiments are thorough, including different models and tasks."
- **Reviewer DxNU**:  “Experimental results indicate success with better detection and improved performance.”

**[Writing]**

- **Reviewer DxNU**: "The custom temperature per attention head is a simple but affective approach."
- **Reviewer G8PB**: "Paper is clearly written and easy to understand."

During the response period, we carefully try our best to provide feedback and conduct supplementary experiments to all comments from reviewers, *We concisely summarize our responses here*:

**[Presentation Issue]**

- We revised a sentence in the abstract to make it more concise.
- We expanded the title of Table 2，and highlighted "using the attention heads obtained from MMLU" in bold in the Table 3's caption.
- A detailed explanation of this concept has been added in Section 4.5.
- We included the workflow diagram of CoLo (Figure 10) in Appendix F.
- We adjusted the order of (a), (b), and (c) in Figure 4.

**[Experiment Issue]**

- We have added experiments involving the CoLo algorithm on Qwen2.5.
- We have included experiments using prompts to mitigate option bias, further strengthening the motivation for intervention strategies.
- We have introduced experiments on global attention scaling interventions, which support the motivation for targeting specific heads.
- We have conducted experiments on sample size and performance, further determining the sample size required for CoLo.
- We have incorporated an ablation study on the impact of hyperparameters $K_l$ and $K_h$ on performance, providing recommended values.
- We have added experiments examining the improvement in the standard deviation of the model’s option debiasing performance with CoLo.
- We have added experiments that use prompts to increase option bias and construct sample pairs for CoLo.

**[More Explanations]**

> **The studied bias is somewhat narrow and extend CoLo to include a broader range of biases in open-domain settings.**

We fully agree with the suggestion to expand the scope to include a broader range of biases. This is why we are committed to employing inference-based interventions rather than relying on prior-based debiasing methods[1].

However, **option bias remains a crucial issue, and a significant body of research has focused on this topic** [2,3,4,5].
We discovered that **using a prompt-based method can also effectively generate sample pairs instead of swapping in the order of options**. We have conducted preliminary experiments in this direction. This simplified approach is highly conducive to extending the study to a wider variety of biases, particularly in open-domain settings. We are confident that our approach has significant potential for scaling to incorporate additional types of biases.

We hope our detailed responses can address the concerns from reviewers. And your valuable comments have helped us to refine our work.

Best regards and thanks,

Authors of #9076

**Reference**

[1] Zheng C, Zhou H, Meng F, et al. Large language models are not robust multiple choice selectors.

[2] Pezeshkpour P, Hruschka E. Large language models sensitivity to the order of options in multiple-choice questions

[3] Wang H, Zhao S, Qiang Z, et al. Beyond the answers: Reviewing the rationality of multiple choice question answering for the evaluation of large language models

[4] Balepur N, Ravichander A, Rudinger R. Artifacts or Abduction: How Do LLMs Answer Multiple-Choice Questions Without the Question?

[5] Xue et al. Strengthened Symbol Binding Makes Large Language Models Reliable Multiple-Choice Selectors, ACL 2024

---

### Meta-Review · Area_Chair_92pA · 2024-12-23

**Metareview:**

Reviewers of this submission positively commented on a simple and effective method, leading to consistent gains in multiple-choice selection problems for LLMs, both in terms of accuracy (modest improvements) and recall standard deviation. Reviewers also commented positively on the cross-domain generalization insight, suggesting an expanded version of that analysis.

Unfortunately, concern remains regarding the impact of this work due to a lack of theoretical explanation of the motivation of the proposed method, the lack of experiments on larger and more diverse models (despite the authors adding small Qwen 2.5 models during the rebuttal) and the rather narrow problem of option bias multiple choice question, which the authors admit "our method is currently effective solely in mitigating option bias". In addition, a reviewer criticised that not enough consideration was given to alternative ideas for option biasing, which the authors could have answered with another set of experiments.

As a result, at this stage, I do recommend this submission for publication at ICLR.

**Additional Comments On Reviewer Discussion:**

All reviewers and the authors engaged in a healthy and productive rebuttal process, clarifying several questions. Reviewers took the authors' responses into account and re-considered their evaluations, with one reviewer raising their score.

---

### Decision · Program_Chairs · 2025-01-22

Reject